# Acoustic-Based Deep Learning Architectures for Lung Disease Diagnosis: A Comprehensive Overview

**DOI:** 10.3390/diagnostics13101748

**Published:** 2023-05-16

**Authors:** Alyaa Hamel Sfayyih, Ahmad H. Sabry, Shymaa Mohammed Jameel, Nasri Sulaiman, Safanah Mudheher Raafat, Amjad J. Humaidi, Yasir Mahmood Al Kubaiaisi

**Affiliations:** 1Department of Electrical and Electronic Engineering, Faculty of Engineering, University Putra Malaysia, Serdang 43400, Malaysia; alyaahamel@gmail.com (A.H.S.); nasri_sulaiman@upm.edu.my (N.S.); 2Department of Computer Engineering, Al-Nahrain University Al Jadriyah Bridge, Baghdad 64074, Iraq; 3Iraqi Commission for Computers and Informatics, Baghdad 10009, Iraq; 4Department of Control and Systems Engineering, University of Technology, Baghdad 10011, Iraq; 5Department of Sustainability Management, Dubai Academic Health Corporation, Dubai 4545, United Arab Emirates

**Keywords:** acoustic signal analysis, lung sound signals, deep learning, respiratory system, signal analysis, CNN

## Abstract

Lung auscultation has long been used as a valuable medical tool to assess respiratory health and has gotten a lot of attention in recent years, notably following the coronavirus epidemic. Lung auscultation is used to assess a patient’s respiratory role. Modern technological progress has guided the growth of computer-based respiratory speech investigation, a valuable tool for detecting lung abnormalities and diseases. Several recent studies have reviewed this important area, but none are specific to lung sound-based analysis with deep-learning architectures from one side and the provided information was not sufficient for a good understanding of these techniques. This paper gives a complete review of prior deep-learning-based architecture lung sound analysis. Deep-learning-based respiratory sound analysis articles are found in different databases including the Plos, ACM Digital Libraries, Elsevier, PubMed, MDPI, Springer, and IEEE. More than 160 publications were extracted and submitted for assessment. This paper discusses different trends in pathology/lung sound, the common features for classifying lung sounds, several considered datasets, classification methods, signal processing techniques, and some statistical information based on previous study findings. Finally, the assessment concludes with a discussion of potential future improvements and recommendations.

## 1. Introduction

The most promising and popular machine learning technique for disease diagnosis, and in particular for illness identification in general, is the deep learning network. It is not surprising given the dominance of diagnostic imaging in clinical diagnostics and the natural suitability of deep learning algorithms for image and signal pattern recognition.

Deep learning can achieve better prediction accuracy and generalization ability despite requiring more training time and computational resources, which suggests that deep learning has a better learning ability. When compared to conventional machine learning, deep learning can quickly and automatically extract information from an image. Traditional machine learning techniques have difficulty identifying audio and images with comparable properties, while the deep learning approach can manage this with ease [1]. Processing sound signals allows for the quick extraction of important information. Traditional machine learning involves performing learning using instructive characteristics, such as Mel Frequency Cepstral Coefficents (MFCCs). In automatic speech and speaker recognition, MFCCs [2] are characteristics that are frequently used. Deep learning entails learning end-to-end directly from data. Both MFCC and conventional machine learning are accomplished through this. In other words, signal processing is formulated as a learning issue in deep learning. Manyartificial intelligence (AI) engineers share this opinion [3].

Deep neural networks may be trained without supplying lesion-based characteristics on massive databases of lung acoustic signals and spectrogram imagery with no provide lesion-based criteria to recognize lung disease status for patients with greater specificity and sensitivity. The key benefit of adopting this computerized illness detection method is model uniformity (on a single input sample, a model predicts the same values every time), high specificity, dynamic result generation, and high sensitivity. Additionally, because a method may have some responsiveness, accuracy and operating points may be adjusted to match the requirements of various clinical scenarios, for example, a screening system with excellent sensitivity.

This work initially discusses the necessity with an overview and motivations of this study field. Then, the method of the survey includes the commonly considered dataset in the literate, sound-based lung disease classification workflow, deep learning algorithms, preprocessing methods, feature extraction, and a comparison between studies that review lung nodule screening. Next, lung diagnosis methods including general examination, histopathology, and history-based techniques are explored. After that, computer-based diagnosis methods such as the wavelet transform, convolutional neural networks (CNNs), cybernetic technology, computer-assisted diagnosis (CAD), and image processing are also analyzed. Finally, we present a summary of some significant literature results, research gaps in the existing literature, and key aspects of successful deep learning models.

The breathing process of humans is separated into two stages: inspiration and expiration. In order to be inspired, one must breathe air into the lungs. During inspiration, the diaphragm lowers, and its muscles contract. As a result, the volume of the chest hollow expands. The air force within the chest hollow drops. High-pressure oxygenated air from outside the body flows quickly into the lungs. The alveoli are reached by oxygenated air in the lungs. The alveoli walls are thin and are bounded by a blood vessel network. The process of expelling air from the lungs is known as expiration. During expiration, the diaphragm muscles relax and the diaphragm travels upward. Then, the capacity of the chest hollow decreases. The pressure within the chest cavity rises. This causes carbon dioxide to be discharged out of the body. This process can be demonstrated as shown in Figure 1.

### 1.1. An Overview and Motivations

In clinical treatment and instructional duties, medical image classification is critical. The traditional approach, on the other hand, has reached its limit in terms of performance. Furthermore, using them requires a substantial amount of time and effort in terms of extracting and selecting categorization factors. Deep neural networks are a new type of machine learning technology that has shown promise in a wide range of categorization challenges. Notably, CNN dominates with the greatest results for a variety of image categorization tasks. Medical image databases, on the other hand, are difficult to find since categorizing them necessitates extensive expert knowledge.

The primary goal of this paper is to present a comprehensive and easy-to-understand review of the applications of deep learning in medical diagnostics. What is the significance of this? It was discovered that a huge number of scientific papers delve into considerable detail about various deep-learning applications. However, based on the performed survey, few studies provide an overview of deep learning with sound-based classification methods that are compared with this study in Table 1.

Deep learning language can be perplexing to academics that are unfamiliar with the field. This review paper provides a brief and clear introduction to deep learning applications in medical diagnostics, and it contributes to the current body of knowledge to a moderate degree. As criteria for this work, the following research questions are used:How diverse is deep learning’s usefulness in medical diagnosis?Will deep learning ever be able to take the place of doctors?Is deep learning still relevant, or will it be phased out?

### 1.2. Contributions and Review Structure

This paper’s key contributions are as follows:This paper provides a wide assessment of the ideas and characteristics of deep learning being used in the realm of medical lung diagnosis.This paper describes the terms ‘‘breath’’, ‘‘respiratory’’, ‘‘lung sounds’’ ‘‘sound signal analysis’’, and ‘‘acoustic-based classifier’’.This survey presents a classification for diagnosis methods of lung disease in the respiratory system and highlights the use of auscultation systems.A respiratory system sound diagnosis framework is also displayed, which provides a general understanding of the inquiry of respiratory system diagnosis.This work makes a significant addition by presenting a complete assessment of current research on augmentation techniques for background auscultation of the respiratory system.This review highlights the role of deep learning CNN integration in enhancing lung auscultation screening.It also makes numerous recommendations for future study opportunities.

This article is separated into three parts. The research technique is given in the first section. Following that, a general idea of deep learning applications in medical diagnostics is given. Finally, the findings are analyzed, conclusions are reached, and recommendations for further research are made.

## 2. Lung Sound Waveforms

### 2.1. The Regular Lung Sound

The regular lung sound waveforms can be divided into:Vesicular breath or normal lung sound: The sound is more high-pitched during inhalation than exhalation, and more intense; it is also continuous, rustling in quality, low-pitched, and soft.Bronchial sound breathing: The sound is high-pitched, hollow, and loud. However, it could be a sign of a health problem if a doctor hears bronchial breaths outside the trachea.Normal tracheal breath sound: It is high-pitched, harsh, and very loud.

A sample of a normal lung sound waveform is shown in Figure 2.

### 2.2. The Wheezing Lung Sound

The wheezing sound is a continuous and high-pitched sound and is distinguished into:Squawks: A squawk is a momentary wheeze that happens while breathing in.Wheezes with numerous notes are called polyphonic wheezes, and they happen during exhalation. The pitch of them may also rise as exhalation nears its conclusion.Monophonic wheezes can last for a long time or happen during both phases of respiration. They can also have a constant or variable frequency.

### 2.3. Crackles Sound

Generally speaking, crackles can be heard while inhaling. They may have a bursting, bubbling, or clicking sound to them.

Coarse: Coarse crackles are louder, lower in pitch, and linger longer in the larger bronchi tubes than fine crackles do. Although they usually occur during inhalation, they can also occur during exhalation.

Medium: These are brought on by mucus bubbling up in the two tiny bronchi, which carry air from the trachea to the lungs. The bronchi are divided into progressively smaller channels that ultimately lead to alveoli, or air sacs.

Fine: These delicate, high-pitched noises are particular to narrow airways. Fine crackles may occur more frequently than coarse crackles during an intake than during an exhalation.

### 2.4. Rhonchi Sound

Low-pitched, continuous noises called rhonchi have a snoring-like quality. Rhonchi can happen when exhaling or when exhaling and inhaling, but not when inhaling only. They take place as a result of fluid and other secretions moving about in the major airways.

### 2.5. Stridorand Pleural Rub Sounds

A high-pitched sound called stridor forms in the upper airway. The sound is caused by air squeezing through a constricted portion of the upper respiratory system.The rubbing and cracking sound known as "pleural rub" is caused by irritated pleural surfaces rubbing against one another.

For efficient respiratory infection therapy, early diagnosis and patient monitoring are critical. In clinical practice, lung auscultation, or paying attention to the patient’s lung sound by means of stethoscopes, is used to diagnose respiratory disorders. Lung sounds are typically characterized as normal or adventitious. The majority of frequent adventitious lung noises heard above the usual signals are crackles, wheezes, and squawks, and their presence typically suggests a pulmonary condition [7,8,9].

The traditional techniques of lung illness diagnosis were detected using an AI-based method [10] or a spirometry examination [11], both of which required photos as input to identify the disorders. Going to a hospital for an initial analysis by X-ray or chest scan in the event of a suspected lung condition, such as an asthma attack or heart attack, is time-consuming, expensive, and sometimes life-threatening. Furthermore, model training with a large number of X-ray images with high quality (HD) is required for autonomising an AI-based system of image-based recognition, which is challenging to obtain each time. A less and simpler resource-intensive system that is able to aid checkup practitioners in making an initial diagnosis is required instead.

In the event of a heart attack, asthma, chronic obstructive pulmonary disease (COPD), and other illnesses, the sounds created through the body’s inner organs vary dramatically. Automated detection of such sounds to identify if a person is in danger of lung sickness is inefficient and self-warning for both the doctor and patients. The technology may be utilized by clinicians to verify the occurrence of a lung ailment. On the contrary, the future extent of this technology consists of integration with smart gadgets and microphones to routinely record people’s noises and so forecast the potential of a case of lung illness.

Nonetheless, the rapid advancement of technology has resulted in a large rise in the volume of measured data, which often renders conventional analysis impractical due to the time required and the high level of medical competence required. Many researchers have offered different artificial intelligence (AI) strategies to automate the categorization of respiratory sound signals to solve this issue. Incorporating machine learning (ML) techniques such as hidden Markov models and support vector machine (SVM) [9] (HMM), CNN, residual networks (ResNet), long short-term memory (LSTM) networks, and recursive neural networks (RNN) are examples of deep learning (DL) architectures [11].

Furthermore, much research has been conducted on feature selection and extraction approaches for automated lung sound analysis and categorization. When performing feature extraction from lung sounds, spectrograms, MFCC, wavelet coefficients, chroma features, and entropy-based features are some of the most typically picked features.

This work reviews the existing deep learning architectures and models to classify adventitious and normal lung sound signals. The rationale behind this is to provide clear insight into using deep learning networks to extract lung disease deep features from input recorded acoustic data to decrease their dimensionality and to achieve handle data imbalance and prediction error reduction.

## 3. Survey Methodology

According to the implemented survey on the diagnosis of lung and respiratory system disease, it is possible to classify the surveyed methods by highlighting the discussed area as shown in Figure 3.

This work discusses the colored topics of the colored blocks in the above figure. We divide the diagnosis methods of the respiratory system into two main methods; clinical and computer-based methods. The clinical assessment methods can be subdivided into three methods: general examination traditional methods, history-based, and histopathology image-based methods. In contrast, computer-based diagnosis techniques can also be subdivided into four common approaches including wavelet, image analysis, image processing, and CNN studies. We highlight CNN-based audio processing as an intriguing field since this technology automatically discovers essential elements without the need for human intervention. The process of excluding and including references in this study is shown in Figure 4, while further detail is illustrated in Table 2.

Simple statistical representation on the most popular considered reference databases is shown in Figure 5, which shows IEEE with 22 (18%), Science Direct and Websites with 14%, MDPI with 12%, and Springer with 9%. The remaining 33% of the selected articles are from other published sources. This indicates that such publications are widely discussed as they touch on human health.

### 3.1. The Commonly Considered Dataset in the Literate

The quality, confidence, and other features of the dataset are essential to measuring the accuracy of training and evaluation of models and architectures that perform on the classification of lung sounds. Several common respiratory/lung sound datasets are listed in Table 3.

### 3.2. Sound-Based Lung Disease Classification Workflow

The major goal of using the respiratory sound dataset is to create a model that can distinguish between healthy and unhealthy lung sounds or to create a model that can distinguish between respiratory disorders by detecting sound abnormalities. The machine learning pipeline employed by the majority of existing projects is divided into three stages. The first is respiratory sound preprocessing using audio filtering and noise-lessening methods. The second phase is feature extraction, which is accomplished by the use of signal processing methods such as spectrum analysis [35,36,37,38], Cepstrum analysis [39,40,41], wavelet transformations [18,42,43], and statistics [44]. The third stage is classification, and the most often used classifiers were K-nearest Neighbors [32,45,46,47,48], Support Vector Machines [49,50,51,52,53], Gaussian Mixture models [54,55], and Artificial Neural Network (ANN) [49,56]. The workflow representation from preprocessing to classification can be shown in Figure 6.

### 3.3. General Methodology Diagram

Unlike transmitted voice sounds, lung sounds are produced inside the lungs and are generated by the larynx. Lung sound signals contain breathing sounds and abnormal or adventitious sounds detected or heard over the chest. Ordinary breathing sounds are listened through the chest trachea or wall. Usually, physicians check chest healthy using the diaphragm of the stethoscope. The normality of breathing sounds is assessed by inspecting the expiration, length of inspiration, symmetry, and intensity of breath sounds. An example of experimental identifying normal and abnormal respiratory audio waveforms is illustrated in Figure 7, where a unique four-channel data 64 collecting device was used to collect respiratory audio waveforms.

The air path inside the chest is disrupted due to a pulmonary deficit. The data enclosed inside the signal waveform, such as strength, timbre (or quality), and frequency are respiratory sound features to help diagnose common lung disorders. The normal breathing sound with its associated noises and distinct features can be classified as shown in Figure 8.

A general framework for deep-learning-based lung sound classification can be shown in Figure 9.

Deep-learning CNNs (DLCNN) are being used to diagnose obstructive lung illnesses, which is a fascinating development. DLCNN algorithms function by identifying patterns in diagnostic test data that are possible for utilization to forecast clinical outcomes or identify obstructive phenotypes. The goal of this study is to present the most recent developments and to speculate on DLCNN’s future potential in the diagnosis of obstructive lung disorders. DLCNN has been effectively employed in automated pulmonary function test interpretation for obstructive lung disease differential diagnosis. For obstructive pattern detection in computed tomography and associated acoustic data, deep learning algorithms such as neural networks using convolutions are state-of-the-art techniques [57]. DLCNN has been applied in small-scale research to improve diagnostic procedures such as telemedicine, lung sound analysis, breath analysis, andforced oscillation tests with promising results.

Deposits in the respiratory system limit airways and induce blood oxygen deficit, resulting in erratic breathing noises. Obtaining these respiratory sounds from test subjects, extracting audio features, and categorizing them will aid in the detection of sputum or other infections. These sickness stages can be accurately classified using deep learning convolution neural network methods. Several studies reviewed DLCNN such as [58], where its goal was to find the best CNN architecture for classifying lung carcinoma based on accuracy and training time calculations. Backpropagation (BP) is the best feed-forward neural network (FFNN) method, with an accuracy of 97.5 percent and training time of 12 s, and the kernel extreme learning machine (KELM) is the best feedback neural network (FBNN) method, with an accuracy of 97.5 percent and an 18 min 04 s training time. A common representation classifier for lung sounds based on deep learning architecture is shown in Figure 10.

### 3.4. Preprocessing

Data preprocessing begins with importing the re-sampling, cropping them, and sound files. Because recordings are made by different research teams using different recording equipment, sampling rates vary (4000 Hz, 44,100 Hz, and 10,000 Hz). All recordings may be re-sampled to a single sampling rate, such as 44,100 Hz, and every sound is typically 3–10 s extended by zero-padding shorter segments and cropping larger ones. The respiratory sound data are divided into distinct breaths during preprocessing by detecting the lack of sound between breaths. Lung sounds captured from different participants will have varying loudness levels. As a result, before processing, the signals were adjusted such that they were roughly the same loudness regardless of the subject. The signals were normalized before being divided into frequency sub-bands using the discrete wavelet transform (DWT). To depict the allocation of wavelet coefficients, a set of numerical characteristics was collected from the sub-bands. A CNN-based scheme was implemented to classify the lung sound signal into one category: squawk, crackle, wheeze, normal, rhonchus, or stridor. The schematic block diagram of the signal preprocessing stage is described in Figure 11.

### 3.5. Deep Learning Algorithms

The majority of studies in the literature used numerous classifiers to see which one produced the greatest accuracy results that are regarded as a main performance metric of study. DLCNN methods such as VGG (VGG-B3, VGG-B1, VGG-V2, VGG-V1,and VGG-D1), Res-Net, LeNet, Inception-Net, and AlexNet were applied to spectrum data for categorization functions, and the results were analyzed and compared with one another to improve categorization of aberrant respiratory sounds. A list of deep learning algorithms that are employed to classify audio signals in literature is demonstrated in Table 4.

### 3.6. Wavelet Transform

A method for extracting and detecting characters based on lung sounds was described in the paper [74]. The wavelet de-noised approach removes noise from the collected lung sounds before employing wavelet decomposition to recover the wavelet features parameters of the de-noised lung sound signals. Because the characteristic vectors of lung sounds have multi-dimensional following wavelet reconstruction and decomposition, a novel technique for converting them into reconstructed signal energy was developed. They also used linear discriminate analysis (LDA) to decrease the length of feature vectors for assessment in order to create a more efficient recognition technique. Finally, they employed a BP neural network to differentiate lung sounds, with 82.5 percent and 92.5 percent recognition accuracy, respectively, using relatively high-dimensional characteristic vectors as the input and low-dimensional vectors as the output. Wavelet Packet Transform (WPT) and classification with an ANN were used to evaluate lung sound data in the study [75]. Lung sound waves were separated into frequency sub-bands using WPT, and statistical parameters were derived from the sub-bands to describe the distribution of wavelet coefficients. The classification of lung sounds as normal, wheezing, or crackling is carried out using an ANN. This classifier was programmed in a microcontroller to construct an automated and portable device for studying and diagnosing respiratory function. In the study [75], a method for distinguishing between two types of lung sounds was provided. The proposed technique was founded on the examination of wavelet packet decomposition (WPD). Data on normal abnormal and normal lung sounds were collected from a variety of patients. Each signal was split into two sections: expiration and inspiration. They used their multi-dimension WPD factors to create compressed and significant energy characteristic vectors, which they then fed into a CNN to recognize lung sound features. Widespread investigational results demonstrate that this characteristic extraction approach has high identification efficiency; nonetheless, it is not yet ready for clinical use.

The role of Wavelet transform can be seen as a part of the de-noising or filtering process as shown in Figure 12.

As the input, a lung sound recording folder is used. Lung sounds are a combination of lung sounds and noise (signal interference).As a signal, sounds can be played and written.The lung sounds are then examined by the scheme, saved in the data, and divided into an array of type bytes.The data array is transformed into a double-sized array.Repeatedly decomposing array data according to the chosen degree of disintegration creates two ranges, every half of the duration of the data range. The initial array is known as a low-pass filter, while the second span is known as a high-pass filter.Wavelet transform is applied to the coefficients in each array.In the data array, both arrays are reconstructed, with a low-pass filter at the beginning and a high-pass filter at the end.The data array is processed via a threshold, creating respiratory sound signal noise and two arrays.Repeat restoration as many times as the stage of restoration is set to each array.In the data array, the order of the preceding half-high-pass filter and half-low-pass filter is reversed, with a discontinuous high-pass filter low-pass filter for every array.Each array’s wavelet transform parameters are re-performed.The data array is then transformed from a double-sized array to a byte-sized array. The acoustic format and folder names that have been specified are functional to the information.A signal (data) of a breathing sound set is restructured to a breathing sound folder, and a data noise array is restructured to a noise beam.

### 3.7. Signal-to-Noise Ratio

The signal-to-noise ratio (*SNR*) is a dimensionless relation of the power of a signal to the associated power noise during recording; this can be expressed by [15]:(1)$$SNR=\frac{P_{signal}}{P_{noise}}={\left(\frac{A_{signal}}{A_{noise}}\right)}^{2}$$
where *A_noise_* denotes root mean square (RMS) of noise amplitude, *A_signal_* represents the RMS of signal amplitude, *P_noise_* denotes the mean of noise power, and the *P_signal_* denotes the mean of signal power.

The optimum wavelet packet foundation for feature extraction was chosen after the space partitioning of wavelet packets [77,78]. They can perform quick random multi-scale WPT and obtain every high-dimension wavelet parameter matrix based on the best basis. The time-domain equal-value relationship between coefficients wavelet and signal energy was then established. The energy was used as an eigenvalue, and vectors of characteristics from a classification ANN were used as forms. This drastically reduces the number of ANN input vectors. Extensive experimental results reveal that the proposed feature extraction approach outperforms other approaches in terms of recognition performance. The time-domain equal-value relationship between wavelet coefficients and signal energy was then established. The energy was used as an eigenvalue, and feature vectors from a classification ANN were used as forms. The number of ANN input vectors are considerably reduced as a result. Extensive experimental findings show that in terms of recognition performance, the suggested feature extraction technique surpasses alternative approaches.

### 3.8. Extracting Features

MFCC was employed as sound clip characteristics [79,80]. Speech recognition systems frequently employ MFCCs [81]. They have also been extensively employed in prior employment on the recognition of unexpected respiratory sound signals because they give an indication of the time domain short-term power spectrum of the sounds. Because multiple adventitious sounds might appear in the same tape at different periods and have varied durations, both the frequency and time content are significant in distinguishing between them. As a result, MFCC is useful for recording a signal’s transform in frequency components during the time. Frequencies are allocated to MEL scales that are nonlinear frequencies with equal distance in the human auditory system. Before further processing, MFCC generates a two-dimensional vector feature (frequency and time) that is compressed into an array of one-dimensional scale. The MFCC computation technique is depicted in Figure 13, while Figure 14 shows a model production of the MFCC content of sounds featuring various adventitious noises.

## 4. Lung Sound Characteristics and Types

During a chest examination, sound signals released by the lung are heard from the chest wall throughout the process of inhalation and exhalation. Because these signals are not fixed, the frequency component fluctuates over time [83,84]. In general, there are three types of respiratory noises: normal (normal) sounds, episodic sounds (wheezing and snoring), and aberrant sounds (chronic diseases such as asthma and lung fibrosis). Figure 15 shows an illustration of the breathing process and its frequency ranges.

### Studies Review Lung Nodule Screening

The studies [57,85,86] provided a survey of cutting-edge deep-learning-based respiratory nodule analysis and screening algorithms, with an emphasis on their presentation and medical applications. The study [85] compared the network performance, limitations, and potential trends of lung nodule investigation. The review [86] evaluated why molecular and cellular processes are of relevance. DLCNN has been used in different diagnostic procedures such as lung sound analysis, forced oscillation test, telemedicine, and breath analysis, with encouraging outcomes in small-scale investigations, according to [57].

In the same context, the papers [84,87,88,89,90,91] reviewed cancer diagnosis of the lung using medical picture analysis. Lung cancer is the foremost source of mortality globally, with “1.76 million related deaths recorded in 2018,” according to [89]. In addition, “Lung has the highest incidence rate of cancer death in both men and women, accounting for over a quarter of all cancer fatalities globally” [90].

Many published journal papers review and proposed original methods to assess lung disease using deep learning CNN as an artificial intelligence technique. For highlighting the importance of these publications, this review briefly provides a table that lists the analyzed sample, the CNN algorithm type, tested data (sound or image samples), and their significant findings as seen in Table 5.

The table shows a classification of some published articles and their achievements. The studies [111,112,113,114] created a problem-based architecture that saves image data for identifying consolidation in a Pediatric Chest X-ray dataset. They designed a three-step pre-processing strategy to improve model generalization. An occlusion test is used to display model outputs and identify the observed relevant area in order to check the reliability of numerical findings. To test the universality of the proposed model, a different dataset is employed as additional validation. In real-world practice, the provided models can be used as computer-aided diagnosis tools. They thoroughly analyzed the datasets and prior studies based on them, concluding that the results could be misleading if certain precautions are not followed.

ANNs are widely used for sentiment analysis, regression issues, and classification. ANNs are a developing field with numerous subtypes. They differ due to a number of factors, including data flow, density, network architecture, and complexity. However, all of the many types aim to mimic and simulate neuronal function in order to enhance deep learning models.

The properties of the frequency and amplitude affect how the sound signal is analyzed.The visual representation of the audio signal’s frequency spectrum is called a spectrogram. These spectrograms are widely employed in a variety of fields, including voice recognition, linguistic analysis, audio classification, and signal processing. There are many potential applications for combining deep learning with audio signal processing. One may use deep learning models to interpret and evaluate the data by turning an audio file’s raw waveform into spectrograms.

A binary classification is typically used in audio classification so that it is feasible to tell if the input signal is the desired audio or not. With the use of a deep learning framework such as TensorFlow [115], the incoming noise signal aggregated during data gathering is transformed into a waveform that can be used for additional processing and analysis. After the waveform has been successfully acquired, it is possible to turn it into a spectrogram, which is a visual representation of the waveform that is available. A deep learning CNN can be used to assess these spectrograms appropriately by building a deep learning model to produce a binary classification result because these spectrograms are visual images.

## 5. Existing Literature Gaps

Some difficulties encountered by researchers during acoustic signal analysis and identification include; (1) previous analysis approaches, particularly non-CNN based deep learning, require very sophisticated analysis architectures that are source-intensive [116]. This suggests that it needs high-end calculation capability, which may involve significant infrastructure expenses. If no infrastructure investment is made, illness prediction and training might consume an extremely long period. Existing approaches, such as physical identification by a physician, also require a lengthy time and multiple hospital appointments to determine if a patient has COPD or not. (2) In many circumstances, the amount of respiratory audio samples is uneven in terms of illness [117]. There is a constant requirement to balance the database as every architecture trained on imbalanced data can forecast the disease with the most samples. (3) The respiratory acoustic samples often contain numerous noises [118]. It should be attended to in a diversity of ways. Key characteristics of effective deep learning models can be listed as follows:Dataset selection: because the whole model is built on it, obtaining and maintaining a noise-free database is crucial. The training data must be properly preprocessed.Algorithm choice: it is significant to grasp the study’s function. A variety of algorithms may be tried to see which ones produce results that are closest to the objective.Feature extraction strategies: it is also an important task in the development of successful models. It is effective when high model accuracy is required, as well as optimum feature selection, which aids in the creation of redundant data throughout each data analysis cycle.

## 6. Conclusions

Deep learning convolutional neural networks (DLCNN) are being used to diagnose obstructive lung illnesses, which is a fascinating development. DLCNN algorithms function by identifying patterns in diagnostic test data that can be applied to forecast and identify obstructive phenotypes or clinical outcomes. DLCNN will require consensus examination, data analysis, and interpretation techniques as it matures as medical technology. To enable big clinical trials and, ultimately, minimize ordinary clinical use, such tools are required to compare, understand, and reproduce study findings from and among diverse research organizations. It is necessary to make recommendations on how DLCNN data might be used to generate diagnoses and influence clinical decision-making and therapeutic planning. This review looks at how deep learning can be used in medical diagnosis. A thorough assessment of several scientific publications in the field of deep neural network applications in medicine was conducted. More than 200 research publications were discovered, with 77 of them being presented in greater detail as a result of various selection techniques. Overall, the use of a DLCNN in the detection of obstructive lung disorders has yielded promising results. Large-scale investigations, on the other hand, are still needed to validate present findings and increase their acceptance by the medical community. We anticipate that physicians and researchers working with DLCNN, as well as industrial producers of this technology, will find this material beneficial.

## Figures and Tables

**Figure 1 diagnostics-13-01748-f001:**
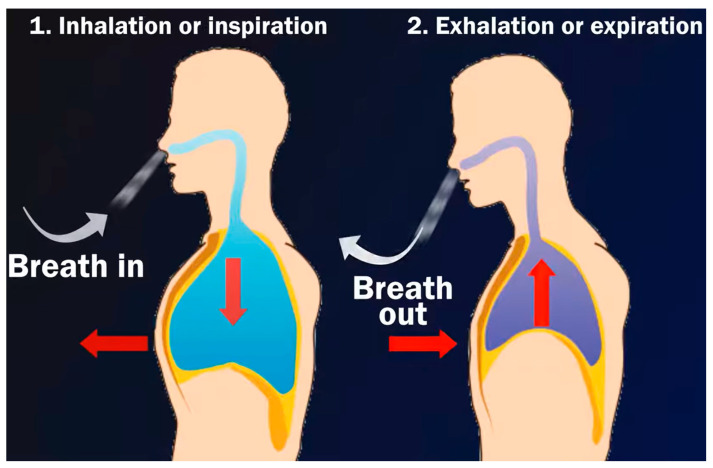
Diaphragm muscles during inhalation and exhalation.

**Figure 2 diagnostics-13-01748-f002:**
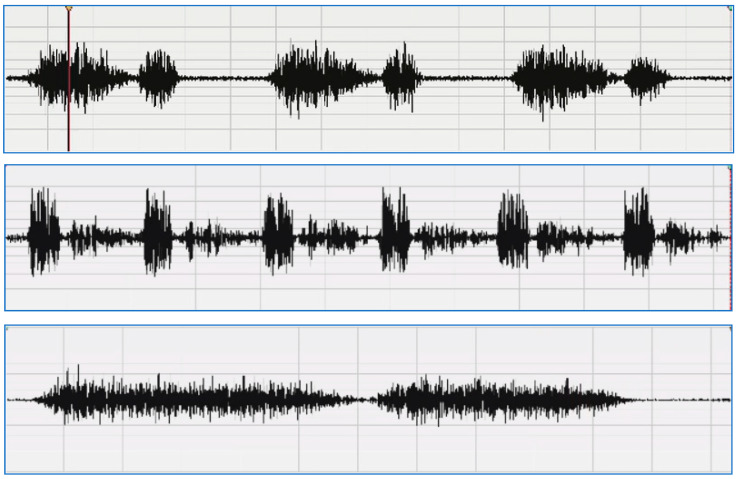
Sample of a normal lung sound waveform [6]: vesicular—normal (**upper**), bronchial (**middle**), normal tracheal (**lower**).

**Figure 3 diagnostics-13-01748-f003:**
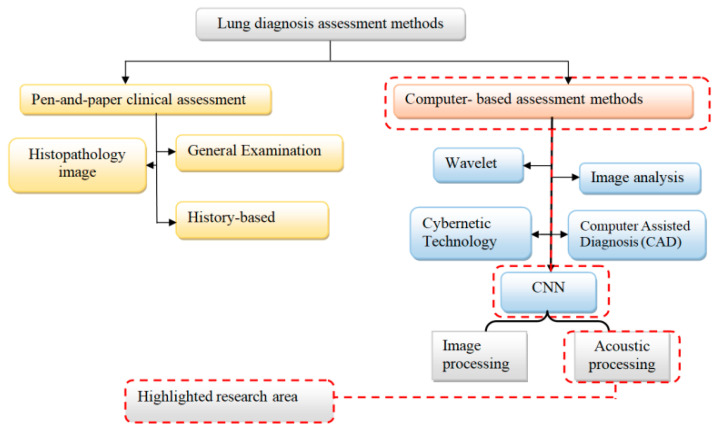
Classification of lung diagnosis methods highlights the focused area in this work.

**Figure 4 diagnostics-13-01748-f004:**
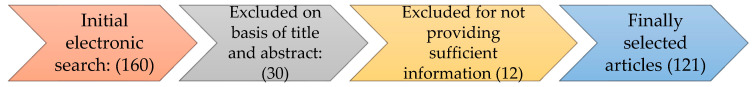
Flow chart of reference selection criteria.

**Figure 5 diagnostics-13-01748-f005:**
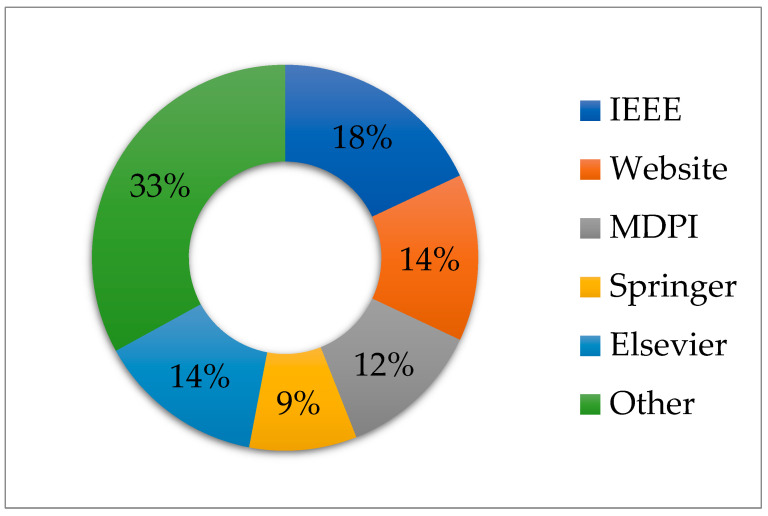
Graphical representation of the databases for the considered references.

**Figure 6 diagnostics-13-01748-f006:**
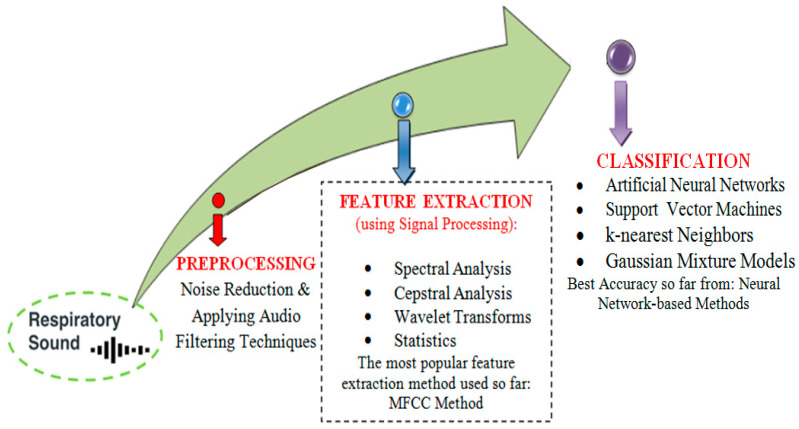
Workflow from preprocessing to classification.

**Figure 7 diagnostics-13-01748-f007:**
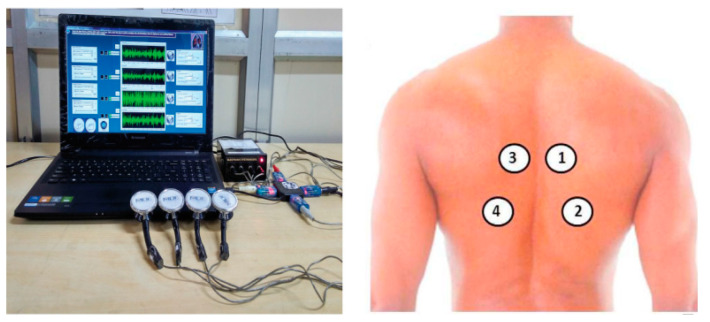
A device for collecting lung sound signals in the back places of the chest. The numbers in the right image correspond to the sensors on the left one.

**Figure 8 diagnostics-13-01748-f008:**
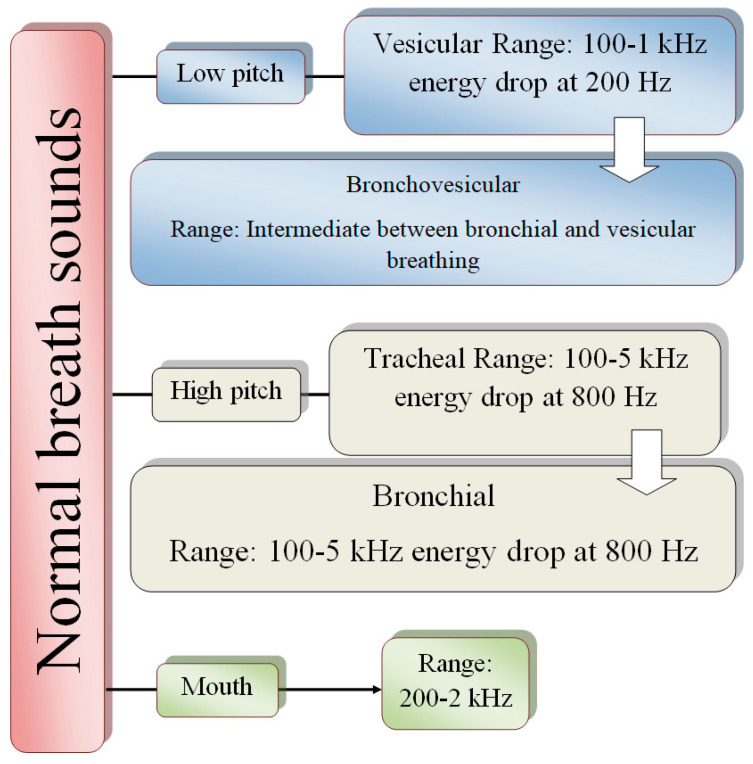
Frequency and power of sound properties.

**Figure 9 diagnostics-13-01748-f009:**
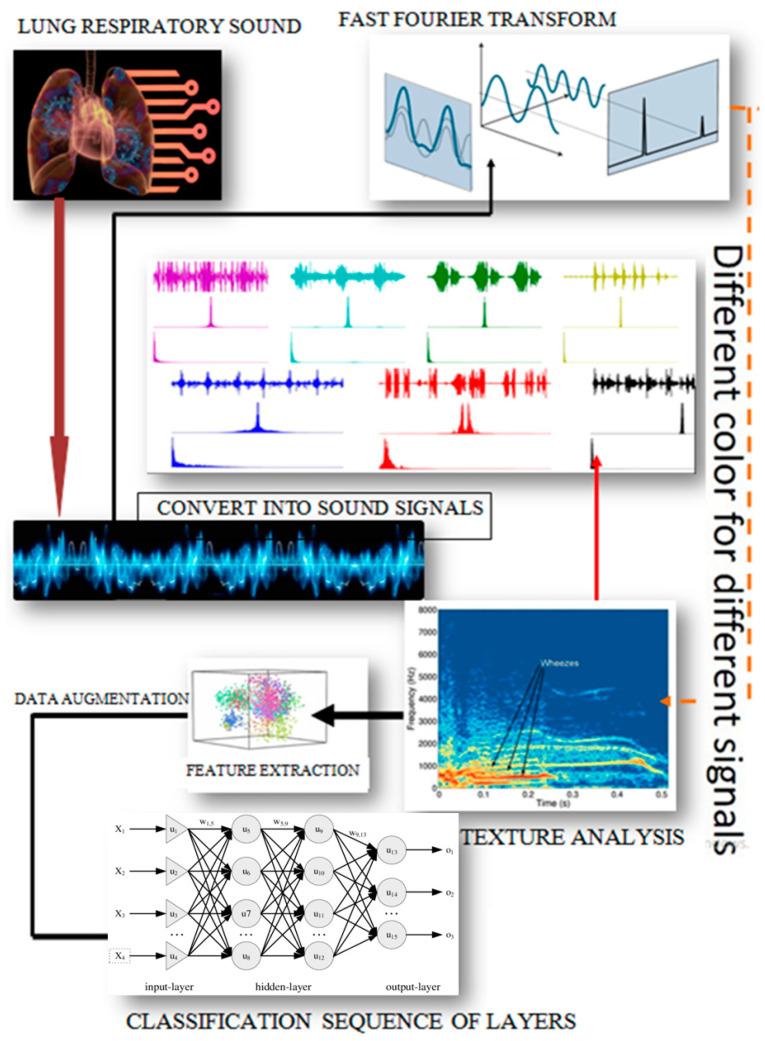
General deep-learning-based lung sound system workflow.

**Figure 10 diagnostics-13-01748-f010:**
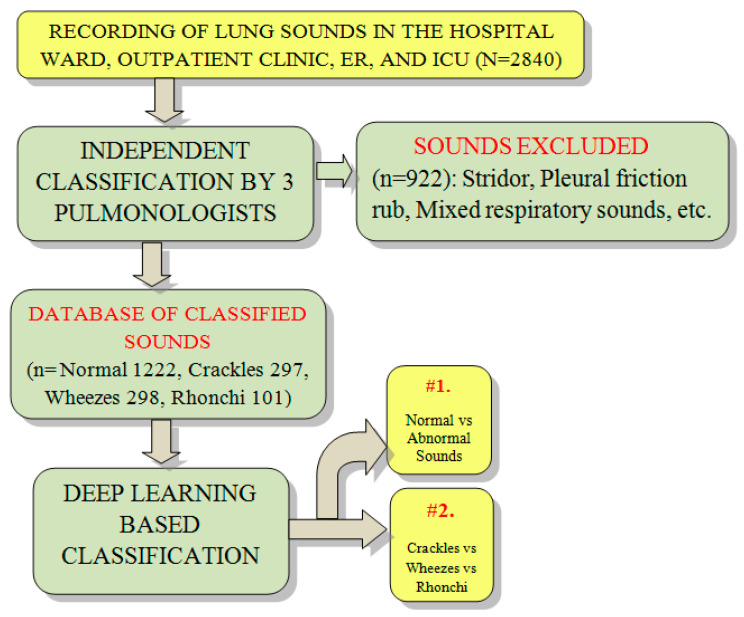
Lung sound classifier based on deep learning architecture [59].

**Figure 11 diagnostics-13-01748-f011:**
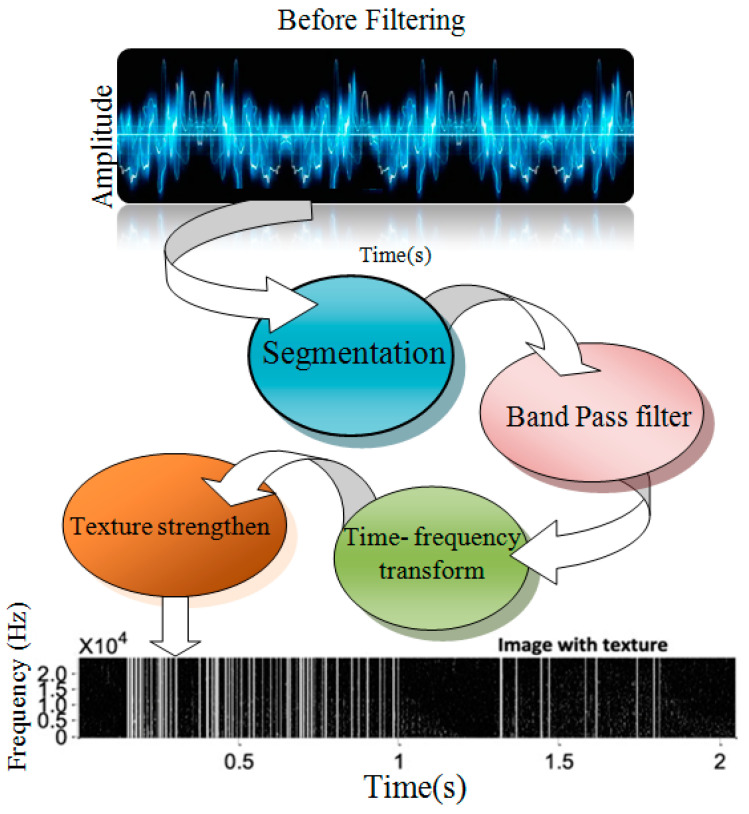
Block diagram of the signal preprocessing stage [60].

**Figure 12 diagnostics-13-01748-f012:**
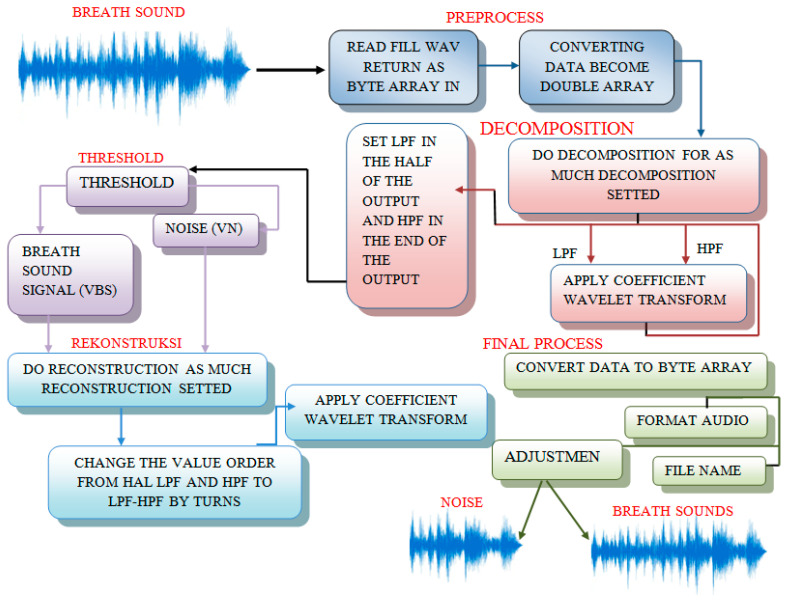
Wavelet transforms as a part of the de-noising or filtering process [76].

**Figure 13 diagnostics-13-01748-f013:**
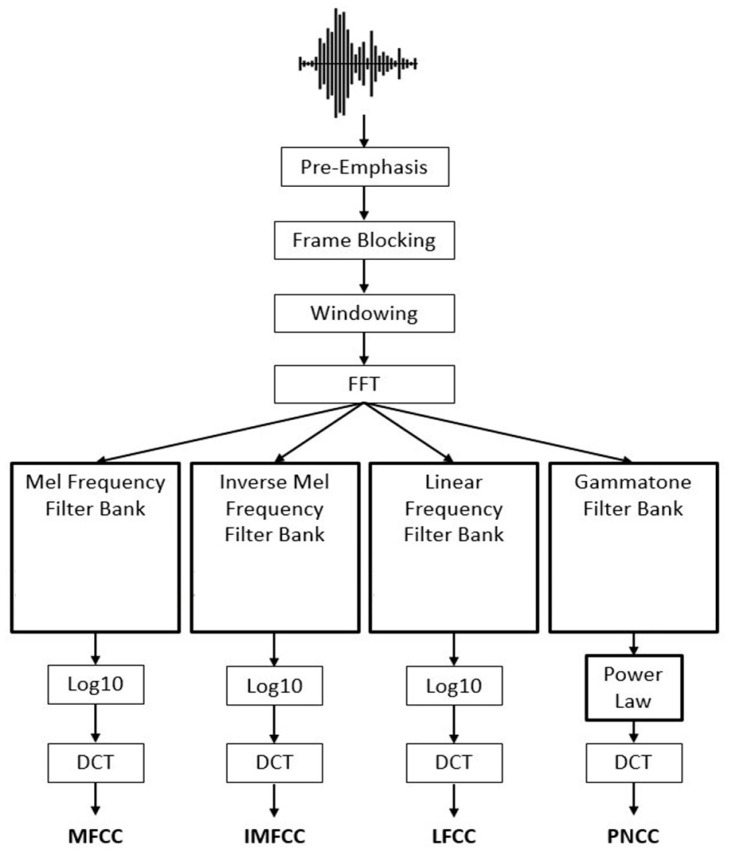
MFCC computation technique [82].

**Figure 14 diagnostics-13-01748-f014:**
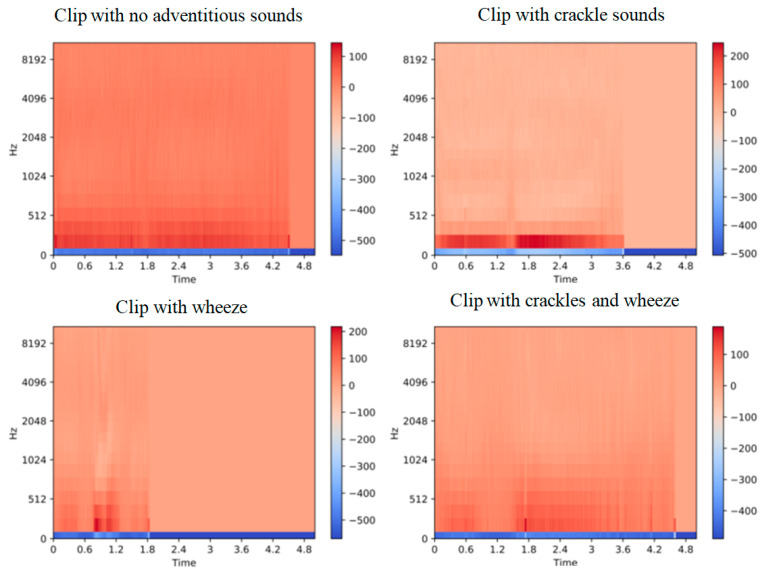
Output sample of MFCC of sounds chosen from each label [31].

**Figure 15 diagnostics-13-01748-f015:**
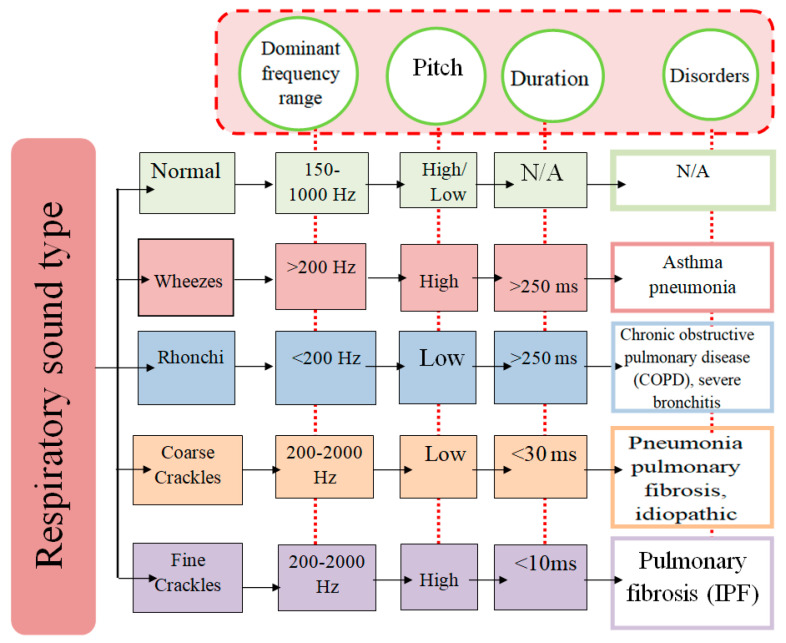
Characteristics of lung sound.

**Table 1 diagnostics-13-01748-t001:** Comparison with related review studies on sound-based respiratory disease classification/detection using deep learning algorithms.

Ref.	Discussion Lung Diseases	Literature Focus	Images/Sound	Highlighting Literature Gaps	Proposing a Solution	Published Date
[4]	Limited	Emerging artificial intelligence methods with respiratory healthcare domain applications	Images+ sound	No	No	2022
[5]	Limited to COVID-19	Deep learning for cough audio sample-based COVID-19 diagnosis	Sound	Yes	Yes	2022
This work	Extensive with most common diseases	Lung disease recognition based on sound signal analysis with machine learning	Images + sound	Yes	Yes	2023

**Table 2 diagnostics-13-01748-t002:** The considered search query of the selected publications.

Search Query	Database	Initial Search	After Remove Repeated Ones	Exclude Based on Title, Abstract	Not Providing Sufficient Info.	Final Selection
Title includes (audio, sound or acoustic) and (lung and/or respiratory), and (deep learning, machine learning, artificial intelligence)	IEEE Xplore	45	32	4	0	28
	Web of science	58	49	12	2	35
Title, abstract, and keywords include (audio, sound or acoustic) and (lung and/or respiratory), and (deep leaning, machine learning, artificial intelligence)	Scopus	76	79	14	7	58
Total		179	160	30	9	121

**Table 3 diagnostics-13-01748-t003:** Common respiratory/lung sound datasets in the literature.

Dataset Name	Description	Used by	Source
Respiratory Sounds Dataset (RSD) ICBHI 2017	Regular sound signals in addition to three kinds of adventitious respiratory sound signals: wheezes, crackles, and a combination between wheezes and crackles.	[12,13,14,15,16,17,18,19]	[20]
HF_Lung_V1	Comprises 9765 lung sound audio files (each lasting 15 s), 18,349 exhalation labels, 34,095 inhalation labels, 15,600 irregular adventitious sounds’ classes, and 13,883 regular adventitious sound classes (including, 4740 rhonchus classes, 8458 wheeze classes, and 686 stridor classes).	[21]	[21]
Respiratory-Database@TR	Each patient has 12-channel lung sounds. Short-term recordings, multi-channel analysis, 5 COPD (chronic obstructive lung disease) severity levels (COPD4, COPD3, COPD2, COPD1, COPD0) (At least 17 s).	[22]	[23]
Own Generated Database	The lung sounds were captured using an e-stethoscope and an amplifier linked to a laptop. An e-stethoscope with a chest piece that is touched by the patient and a microphone-based recording sound signals with a 44,100 Hz sampling rate that is attached to signal amplifiers are used in this setup. The amplifier kits extend the signal range to about (70–2000 Hz) with respiratory sounds (with frequency controller and control amplifier) when associated with an earphone (to listen to live records) and a PC.	[24]	[24]
Own Generated Database	Data are separated into two types: sub-interval set, which includes complete patient set, which comprises all patients’ measures and is classed as abnormal or normal, counting all patients’ sub-interval measurements of any duration. It has around 255 h of measured lung sound signals.	[25]	[25]
Own Generated Database	RSs non-stationary data collection with 28 separate patient records. For training and testing, two distinct sets of signals were employed. Except for crackles and wheezes, which were data from six patients each, each class in the training and test sets comprised two recordings from distinct patients. The sampling frequency of the recorded data was 44.1 kHz.	[26]	[26]
R.A.L.E. Repository	It is a collection of digital recordings of respiratory sounds in health and sickness. These are the breath sounds that physicians, nurses, respiratory therapists, and physical therapists hear using a stethoscope when they auscultate a patient’s chest. Try-R.A.L.E. Lung Sounds, which provides a vast collection of sound recordings and case presentations, as well as a quiz for self-assessment.	[27]	[28]
R.A.L.E. Lung Sounds 3.0	It includes five regular breathing recordings, four crackling recordings, and four wheeze recordings. To eliminate DC components, a first-order Butterworth high-pass filter with a cut-off frequency of 7.5 Hz was employed, followed by an eighth-order Butterworth low-pass filter with a cut-off frequency of 2.5 kHz to band restrict the signal.	[29]	[30]
Respiratory Sound Database	It was developed by two Portuguese and Greek research teams. It has 920 recordings. The duration of each recording varies. 126 patients were recorded, and each tape is documented. Annotations include the start and finish timings of each respiratory cycle, as well as if the cycle comprises wheeze and/or crackle. Wheezes and crackles are known as adventitious noises, and their presence is utilized by doctors to diagnose respiratory disorders.	[12,18,24,31,32,33]	[34]

R.A.L.E. refers to (Respiratory Acoustics Laboratory Environment).

**Table 4 diagnostics-13-01748-t004:** Deep learning algorithms for audio classification.

Network	Ref.	Acronym	Year	Other Variants
VGG	[61,62]	Visual Geometry Group	2014	VGG-D1, VGG-V2, VGG-V1, VGG-B3, and VGG-B1
Alex-Net	[63]	Krizhevsky, Alex	2012	Its architecture has sixty million parameters
ResNet	[64]	Residual Neural Networks	2015	An error rate of 3.6 percent
Inception Net	[65,66,67,68]	InceptionNet or GoogleNet	2014	With a 6.67 percent error rate, four million parameters
LeNet	[69,70,71]	Yann LeCun et al.	1998	It contains a full link layer, pooling layer, and convolutional layer.
M-CNN	[72]	Multi-scale CNN	2017	Several convolutional layers are stacked over the real vector to extract the higher-level features.
ML-CNN	[73]	Deep-learning-based disease NER architecture (ML-CNN)	2017	Lexicon feature, character-level, and word level. Embeddings are concatenated as input of the CNN model

**Table 5 diagnostics-13-01748-t005:** List of the analyzed sample, the CNN algorithm type, tested data (sound or image samples), and their significant findings for the publications that have been surveyed.

Study	Method	Splitting Strategy	Performance			
			Specificity	Sensitivity	Accuracy	Score
Demir et al. [92]	VGG16	10-fold CV	-	-	63.09%	-
Serbes et al. [93]	SVM	official 60/40	-	-	49.86%	-
Sen I, et al. [94]	GMM Classifier	-	90%	90%	85.00%	-
Saraiva et al. [95]	CNN	random 70/30	-	-	74.3%	-
Yang et al. [96]	ResNet + SE + SA	official 60/40	81.25%	17.84%	-	49.55%
Ma et al. [97]	bi-ResNet	official 60/40 random 10-fold CV	69.20% 80.06%	31.12% 58.54%	52.79% 67.44%	50.16% 69.30%
Pham et al. [98]	CNN-MoE	official 60/40 random 5-fold CV	68% 90%	26% 68%	-	47% 97%
Gairola et al. [99] official 60/40	CNN	official 60/40 interpatient 80/20	72.3% 83.3%	40.1% 53.7%	-	56.2% 68.5%
Liu et al. [100]	CNN	random 75/25	-	-	81.62%	-
Acharya and Basu [101]	CNN-RNN	interpatient 80/20	84.14%	48.63%	-	66.38%
Allahwardiand Altan et al. [102]	Deep Belief Networks (DBN)	-	93.65% 73.33%	93.34% 67.22%	95.84% 70.28%	
Kochetov et al. [103]	RNN	interpatient 5-fold CV	73%	58.4%	-	65.7%
Minami et al. [104]	CNN	official 60/40	81%	28%	-	54%
Georgios Petmezas et al. [12]	CNN-LSTM with FL	Interpatient 10-fold CV LOOCV	84.26% -	52.78% 60.29%	76.39% 74.57%	68.52% -
Chambres et al. [105]	HMM SVM	official 60/40	56.69% 77.80%	42.32% 48.90%	49.50% 49.98%	39.37% 49.86%
Oweis et al. [26]	ANN	-	100%	97.8%	98.3%	-
Jakovljevi’c and Lonˇcar-Turukalo [106]	HMM	official 60/40	-	-	-	39.56%
Bahoura [27]	GMM	-	92.8%	43.7%	80.00%	-
Emmanouilidou D et al. [25]	RBF SVM Classifier	-	86.55 (±0.36)	86.82 (±0.42)	86.70%	-
Ma et al. [107]	ResNet + NL	official 60/40 interpatient 5-fold CV	63.20% 64.73%	41.32% 63.69%	-	64.21% 52.26%
Nangia et al. [24]	CNN	-	-	-	94.24%	93.6%
Pramono RX et al. [29]	SVM	-	83.86%	82.06%	87.18%	82.67%
Nguyen and Pernkopf [108]	ResNet	official 60/40 official 60/40	79.34% 82.46%	47.37% 37.24%	- 73.69%	58.29% 64.92%
Bardou D et al. [7]	CNN	-	-	-	95.56%	-
Aykanat M et al. [109]	ANN	-	86%	86%	76.00%	-
Chamberlain et al. [110]	-	-	0.56	-	86% Wheeze	-

## Data Availability

All data underlying this article will be shared on reasonable request to the corresponding author.

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
