# Peer review of "Acoustic-Based Deep Learning Architectures for Lung Disease Diagnosis: A Comprehensive Overview"

_diagnostics, 2023, doi:10.3390/diagnostics13101748_

Round 1

Reviewer 1 Report

This review was not well organized which was not thoroughly informative.

Too much entry level knowledge was on breathing.

Resolution of figures should be improved.

Please check the copyright of figure 1.

The authors may want to be cautious to claim that “few studies provide a comprehensive overview of deep learning's applicability in medical diagnosis”.

Section 2.1 may be better to be summarized as a table.

Please clarify the benefit of using deep learning method compared to the conventional signal processing or machine learning.

The authors may need to introduce the common preprocessing methods before the data was into the deep learning model.

How the structures of network model vary, and progress should be analyzed.

The waveform of respiratory sound with different diagnosis features can be provided.

Reviewer 2 Report

The article entitled “Acoustic-based Deep learning architectures for lung disease diagnosis” is well written and well structured and, from my point of view, would be of interest for the readers of Diagnostics. In spite of this and before its publication, I would recommend authors to perform the following changes:

Introduction: from my point of view, before starting with the content of line 31 it would be of interest introducing a paragraph about Deep neural networks.

I am quite surprised of not finding any bibliographical reference from line 41 to 91. My background is in machine learning but I have great concert that referencing is required for those paragraphs.

Please also note that the details about the study design that starts in line 58 should be moved to Materials and Methods.

Line 228: consider explaining the meaning of the acronym RALE.

Line 418: Mel Frequency Cepstral Coefficient is introduced. Please include any bibliographical reference to this coefficient.

In section 4 it speaks about existing literatura gaps but no bibliographical refences are cited. From my point of view, it would be of interest for the readers that some examples of literature be included in this section in order to make more easy to understand the suggested gaps.

Reviewer 3 Report

The authors present a review paper on acoustic-based deep learning architectures for lung disease diagnosis. However, they also refer other works based on image processing/classification for lung disease diagnosis, and they do not sufficiently clarify the differences between these works and the ultimate goal of the work presented in this manuscript.

As such, my first recommendation is that the title (and keywords) should reflect the fact that this is a review paper.

The paper also needs a complete English revision; please refer to the attached commented PDF document where some of the needed corrections are highlighted.

The authors should define the acronyms/abbreviations the 1st time they use it; also refer to the attached commented PDF document where some of the cases are highlighted.

Always use formal English language.

The manuscript also need a structural improvement; please also refer to the attached commented PDF document where these improvements are noted.

Section “3. Survey Methodology” should also include and explain the search query (the one that gives 160 papers has the result), and, additionally, the used databases should also be mentioned (and the numbers of papers retrieved for each database, number of repeated papers, etc.).

Some other minor corrections/improvements are also noted in the attached commented PDF document; please address them.

Author Response

Initially, I would like to thank the editor and the reviewers for their effort and for giving us the opportunity to revise and resubmit our manuscript.

Since the entire reviewer's comments are marked directly on the manuscript document, we have considered all these comments directly on the manuscript with tracking change version.

We think this revision has improved the manuscript and hope you find it worthy of publication.

Round 2

Reviewer 1 Report

The authors have given comprehensive response to the reviewer, and the reviewer agreed to accept it. 

Author Response

Thank you for the effort and for giving us the opportunity to revise and resubmit our manuscript.

Reviewer 3 Report

As mentioned before, section “3. Survey Methodology” should also include and explain the search query (the one that gives 160 papers has the result), present and explain the rules applied in step two of the process, and present the require information in step three (following the steps in figure 4); please also refer to the attached commented PDF document where these improvements are noted.

All my other previous concerns and questions have been addressed.

Author Response

All the reviewer comments are considered in the new version of the manuscript.

The attached file includes the revised version highlighting the change.  
